# Bacterial Metabarcoding of Raw Palm Sap Samples from Bangladesh with Nanopore Sequencing

**DOI:** 10.3390/foods13091285

**Published:** 2024-04-23

**Authors:** Ágota Ábrahám, Md. Nurul Islam, Zoltán Gazdag, Shahneaz Ali Khan, Sharmin Chowdhury, Gábor Kemenesi, Sazeda Akter

**Affiliations:** 1National Laboratory of Virology, Szentágothai János Research Centre, University of Pécs, 7624 Pécs, Hungary; agotaabraham@gmail.com; 2Department of Forest and Wildlife Ecology, University of Wisconsin—Madison, Madison, WI 53705, USA; nurul.dvm@gmail.com; 3Institute of Biology, Faculty of Sciences, University of Pécs, 7622 Pécs, Hungary; gazdag@gamma.ttk.pte.hu; 4Department of Physiology Biochemistry and Pharmacology, Chattogram Veterinary and Animal Sciences University, Chattogram 4225, Bangladesh; shahneazbat@gmail.com; 5Department of Pathology and Parasitology, Faculty of Veterinary Medicine, One Health Institute, Chattogram 4202, Bangladesh; sharminchowdhury@cvasu.ac.bd; 6Department of Medicine and Surgery, Faculty of Veterinary Medicine, Chattogram Veterinary and Animal Sciences University, Chattogram 4225, Bangladesh; sazeda@cvasu.ac.bd

**Keywords:** NGS, food safety, mobile lab, bacteria, microbial community

## Abstract

The traditional practice of harvesting and processing raw date palm sap is not only culturally significant but also provides an essential nutritional source in South Asia. However, the potential for bacterial or viral contamination from animals and environmental sources during its collection remains a serious and insufficiently studied risk. Implementing improved food safety measures and collection techniques could mitigate the risk of these infections. Additionally, the adoption of advanced food analytical methods offers the potential to identify pathogens and uncover the natural bacterial diversity of these products. The advancement of next-generation sequencing (NGS) technologies, particularly nanopore sequencing, offers a rapid and highly mobile solution. In this study, we employed nanopore sequencing for the bacterial metabarcoding of a set of raw date palm sap samples collected without protective coverage against animals in Bangladesh in 2021. We identified several bacterial species with importance in the natural fermentation of the product and demonstrated the feasibility of this NGS method in the surveillance of raw palm sap products. We revealed two fermentation directions dominated by either *Leuconostoc* species or *Lactococcus* species in these products at the first 6 h from harvest, along with opportunistic human pathogens in the background, represented with lower abundance. Plant pathogens, bacteria with the potential for opportunistic human infection and the sequences of the *Exiguobacterium* genus are also described, and their potential role is discussed. In this study, we demonstrate the potential of mobile laboratory solutions for food safety purposes in low-resource areas.

## 1. Introduction

Date palm sap collection is a widespread and common practice during the winter season in South Asia. Most commonly, it is consumed raw or processed into other products, such as molasses or tari. There are several food safety concerns during these processes, from collection to storage and transport and later processes. Nipah virus is a well-known and highly lethal viral pathogen in relation to this food product. All Nipah cases in Bangladesh to date have been epidemiologically linked to raw date palm sap consumption. These index cases presumably involved the consumption of sap contaminated by flying foxes (*Pteropus* bats) during collection [1]. Flying foxes are represented by 51 species, mostly occurring in Southeast Asia, South Asia and Oceania. There are widespread habitat losses and hunting activities against these animals in most places within their distribution area. In South Asia, mostly due to habitat losses and limited food sources, they are frequent visitors on agriculture-related fruits and sugar sources (such as palm sap). Due to these contacts, they are considered a priority species in relation to outbreak prevention strategies [2]. Although the consumption of this crop has known health risks associated with infectious diseases, it is still a widely common practice. In a survey-based study conducted in Bangladesh, more than one third of the responders reported drinking raw palm sap at least once per month. Ten percent of the responders related the consumption of unprocessed palm sap to mild diseases, such as diarrhea, vomiting or indigestion [1,3]. Nevertheless, the bacterial components of this product are less studied and there is not enough knowledge about the fermentation processes or common pathogenic bacterial species and their origins. 

Considering the importance of this product as a nutrient during the winter season, increased food safety awareness is required both from general consumers and from the respective authorities. Novel techniques may help to conduct the non-targeted and rapid surveillance of pathogens. Bacterial metabarcoding offers a solution for the discovery and the examination of bacterial communities.

Oxford Nanopore Technologies (ONT) stands out as the sole platform currently facilitating off-lab and in-field (point-of-care) library preparation and sequencing, coupled with online real-time data analysis [4]. This feature is particularly advantageous for diagnostic applications conducted in the field. Moreover, ONT is an emerging technology in food safety applications, where it has already been developed for different food-borne pathogen screening applications, with reduced time, materials and expenses needed compared to other technologies [5]. Full-length 16S rRNA sequencing is suitable for large-scale microbiome studies, such as food safety investigations, and is considered an emerging technology [6].

In our study, we performed a proof-of-concept bacterial metabarcoding analysis on raw date palm sap samples, focusing on the demonstration of the advantages of ONT fast bacterial metabarcoding (full-length 16s rRNA-based) and its mobility. This was carried out as an initial step towards establishing a reference for future food safety interventions with a mobile laboratory setup. Mobile lab solutions offer the significant advantage of enabling on-site, rapid diagnostic testing, thus reducing the response time in public health emergencies and outbreak investigations. By leveraging the portability of nanopore-based sequencing techniques, we were able to conduct this analysis in remote, low-resource locations. The primary aim of this study was to demonstrate that nanopore sequencing-based bacterial metabarcoding is an effective and rapid method for the identification of bacterial communities in food products, offering a significantly shorter diagnostic time-frame compared to conventional methods. With our study, we targeted a common but under-investigated food type, raw date palm sap, usually collected in remote places. The potential of mobile laboratory solutions for food safety purposes has been less explored previously.

## 2. Materials and Methods

### 2.1. Sample Collection

We collected 15 samples in January 2022, in Bangladesh, in the Rajshahi district, Pochamaria, from separate collection pots; this was conducted with the help of a gachi, a local palm sap collector, who admitted to not using protective coverage during the collection of the sap. Therefore, animals and contamination could potentially reach the product. Samples were taken in 50 mL falcon tubes (CLEARLine, Biosigma, Venice, Italy) and were frozen at −20 °C until laboratory processing. The study workflow is summarized in Figure 1.

### 2.2. Sequencing and Analysis

We sedimented 2 mL of each sample with centrifugation for 30 min at 15,000× *g*; thereafter, the sediment was subjected to nucleic acid extraction with the Zymo Quick DNA Miniprep Kit (Zymo Research Corp, Irvine, CA; USA), following the manufacturer’s protocol. Briefly, the sediment in 200 μL palm sap was lysed in 4× Genomic Lysis Buffer (Zymo Research Corp, Irvine, CA, USA), and, following two rounds of washing steps, the DNA was eluted with 50 μL DNA Elution Buffer (Zymo Research Corp, Irvine, CA, USA).

For quantification during library preparation, we used a Qubit 4 Fluorometer, with the relevant data detailed in the accompanying table. The amplification targeted approximately 1500 bp fragments, encompassing the entire 16S gene, and both this step and the library preparation were conducted according to the rapid sequencing amplicons of the nanopore 16S barcoding protocol (SQK-16S024, Oxford Nanopore Technologies (ONT), Oxford, UK). Both the amplification step and library preparation were performed following the protocol of the nanopore 16S barcoding kit. Briefly, 10 ng-s of genomic DNA was used for the amplification of the 16S gene from each sample. The primers supplied by the company were also tagged with specific barcodes, so, during amplification, the samples were barcoded as well. Amplicons from the PCR products were cleaned with SPRI beads (AmpureXP, Beckman-Coulter, Brea, CA, USA). Samples were pooled together for the rapid adapter (RAP, ONT, Oxford, UK) attachment step, and 100 fmol of the final library was loaded on an R9.4.1. flow cell. The sequencing took place on a MinION Mk1B, lasting 42 h and employing only the fast base calling technique (model: dna_r9.4.1_450bps_fast) with the Guppy base caller (v6.4.8, ONT, Oxford, UK). An analysis was subsequently performed using EPI2ME’s “Fastq 16S” workflow (EPI2ME, ONT, Oxford, UK). 

## 3. Results and Discussion

The study successfully identified several bacterial genera important in the natural fermentation process of raw date palm sap (*Leuconostoc* spp., *Lactococcus* spp.). Moreover, some additional genera were identified with low abundance, with the potential for opportunistic infection in humans (*Citrobacter* spp., *Enterobacter* spp.) (Figure 2). 

This demonstrates the feasibility of nanopore sequencing for the monitoring of such products and also its ability to examine bacterial communities at the genus level. In addition, as a major advantage of the technology, nanopore sequencing produces complete 16S rDNA sequences, resulting in the detailed taxonomic resolution of bacterial communities. 

The carbohydrate concentration of palm sap usually reaches up to 95 g/100 g on a dry matter basis, containing mainly sucrose, glucose and fructose [7]. The research discovered two primary fermentation directions in the freshly collected raw date palm sap, dominated by either *Leuconostoc* species or *Lactococcus* species. With lower representation, only in samples 5 and 15, *Lactococcus* sp. was the dominant species, while, in the other 13 samples, it was *Leuconostoc* sp. (Figure 3). 

*Leuconostoc* spp. are heterofermentative lactic acid bacteria (LAB); they ferment sugars into lactic acid and ethanol/acetic acid. Young *Leuconostoc mesenteroides* cultures produce lactate, acetic acid and dextran from sugars, and the stationary-growth-phase cultures produce lactic acid and ethanol. The *Streptococcus* and *Lactococcus* genera are members of the Lactobacillales order, similarly to the *Leuconostoc* genus, but members of the first two genera are homofermentative LAB, producing only lactate from glucose fermentation [8]. LAB produce bacteriocins, hydrogen peroxide and antimicrobial peptides, in addition to acids and alcohols, which inhibit, among others, the growth of *Escherichia coli (E. coli)*, *Listeria monocytogenes*, *Pseudomonas aeruginosa* and *Staphylococcus aureus* strains [9]. Reuben et al. documented that LAB strains showed antagonistic activity against *E. coli*, *Enterococcus faecalis*, *Salmonella enterica var* Typhimurium and *Salmonella enterica var* Enteritidis [10]. LAB also inhibit the growth of *Citrobacter* [11], *Enterobacter* [12] and *Serratia marcescens* [13].

Fermentation end- and by-products are the reasons that *Streptococcus* spp. only occur with *Lactococcus* spp. In high-sugar-content medium, LAB strains exclude and overgrow species from the Enterobacterales order, which are plant pathogens and common soil microbes. Recycled and unwashed clay pots provided the starter culture, causing some tested samples to appear as “pure” cultures (barcodes 7, 12, 24).

*Acinetobacter*, a common soil bacterium, was the third most frequently detected genus among the 15 samples. A previous study showed that *Acinetobacter* was also present in 4% of grapevine sap samples [14]. Furthermore, another study confirmed the presence of *Acinetobacter* in floral nectar, a medium with similarly high sugar content, in wild Mediterranean plants [15].

Since the dominance of either of these bacteria was obvious in the pre-market stage of collection, we hypothesize the natural fermentation process as the protective state of this product, out-competing potentially harmful bacteria. However, after the consumption of unprocessed palm sap, community members reported a range of mild diseases, possibly related to enteric pathogens. This highlights the importance of rapid and generalized food safety diagnostic measures, as presented in our current study, to discover these pathogens. Moreover, this highlights the potential of opportunistic pathogens with low abundance as the sources of these negative health impacts. We verified the presence of such bacteria; however, their origin is uncertain and more focused research is necessary to better understand the sources of contamination.

Overall, the species composition was similar throughout all 15 samples, as all were dominated by fermenting bacteria and other opportunistic bacterial taxa [16,17,18,19,20]. Since our study focused on the presentation of the nanopore NGS technique for application in food safety practices, our goal was solely to understand the bacterial composition of this food product; therefore, we did not implement any replicates or negative controls. Detailed sequencing results by barcode, with the most abundant genera, are summarized in Appendix A. It is uncertain whether there is any difference between regions of collection or across different seasonal patterns, which may lead to more optimal circumstances for pathogenic bacteria. This also needs to be addressed in future studies.

Beyond fermentation bacteria, we identified a number of potentially opportunistic bacteria from the *Enterobacter*, *Citrobacter*, *Acinetobacter*, *Streptococcus* and *Serratia* genera. *Enterobacter* was found in four samples (barcodes 4, 8, 9, 13); *Citrobacter* in one sample (barcode 6); *Acinetobacter* in seven samples (barcodes 1, 2, 3, 4, 6, 9, 11); and *Serratia* in four samples (barcodes 5, 6, 11, 13). Interestingly, *Streptococcus* was identified only in two samples, where the dominance of *Lactococcus* was observable. However, the sequencing method could not provide species-level details about these. Nonetheless, the presence of these bacteria supports the potential to cause negative health effects, such as gastrointestinal problems. In particular, *Streptococcus* may be associated with fecal contamination, which may originate from visiting animals during collection [21]. These genera have multiple members with opportunistic pathogenic potential, and some of them are known to cause nosocomial infections, such as some Enterobacter species [16]. Our study reports the genera-level taxonomic composition of these materials, but it is challenging to assess the exact pathogenic risk without the species-level identification of the bacteria. Nonetheless, this study serves as an initial step towards assessing the risk of these food products by cataloging the bacterial genera present. Future studies would benefit from assessing the species-level composition and identifying resistance genes to facilitate a more comprehensive risk assessment.

Interestingly, we identified multiple plant pathogenic bacteria, represented by the *Erwinia* and *Pectobacterium* groups, in the data. These are very important to raise awareness in regard to maintaining clean cultivation and harvesting practices to avoid the cross-contamination of palm trees with pathogens during the cutting process [22]. Although these potential plant pathogens were identified in three samples, *Erwinia* in two samples (barcodes 9, 13) and *Pectobacterium* in one sample (barcode 6), the importance of safe agriculture is notable.

Most interestingly, we report the presence of *Exiguobacterium* in one of the palm sap samples, with an over 1% sequence ratio (barcode 9). A range of isolates from diverse environments have been investigated for their potential in biotechnology and industry, such as in producing enzymes, bio-remediation and breaking down harmful substances discharged into the environment. Additionally, certain isolates with the ability to promote plant growth are under study for their capacity to enhance agricultural yields. The role of these bacteria in this food product is uncertain [23].

## 4. Conclusions

Our study served as an initial step towards establishing a reference for future food safety interventions, highlighting the effectiveness of nanopore sequencing in remote locations. The effectiveness of the method includes the high mobility and fine-scale resolution of the results, as demonstrated in this study. Moreover, the possibility for mobile experimental and bioinformatic analysis offers a faster diagnostic time-frame compared to conventional methods. Logistical barriers can delay the timely delivery of samples, affect the acquisition of crucial diagnostic results and hinder the implementation of intervention strategies where time is critical. In our study, we demonstrated an alternative strategy for the implementation of food safety diagnostic practices outside the traditional laboratory setting, using a simple NGS workflow to overcome the diagnostic challenge of unculturable bacteria with classical methods.

The study reinforces earlier findings about the health risks associated with consuming raw date palm sap, particularly highlighting the risk of bacterial contamination. The samples analyzed showed the presence of various opportunistic human pathogenic bacteria, albeit in low abundance. This research underscores the potential health hazards and emphasizes the need for caution and further safety measures in the consumption of this product. Our research also identified the presence of plant pathogenic bacteria in the samples, underscoring the need for clean collection practices. This finding is significant as it highlights the potential adverse impact on agricultural practices, emphasizing the importance of maintaining hygiene standards during the collection of raw date palm sap in relation to plant pathogens.

Overall, the study provides significant insights into the bacterial composition of raw date palm sap and demonstrates the utility of advanced sequencing techniques in food safety surveillance.

## Figures and Tables

**Figure 1 foods-13-01285-f001:**
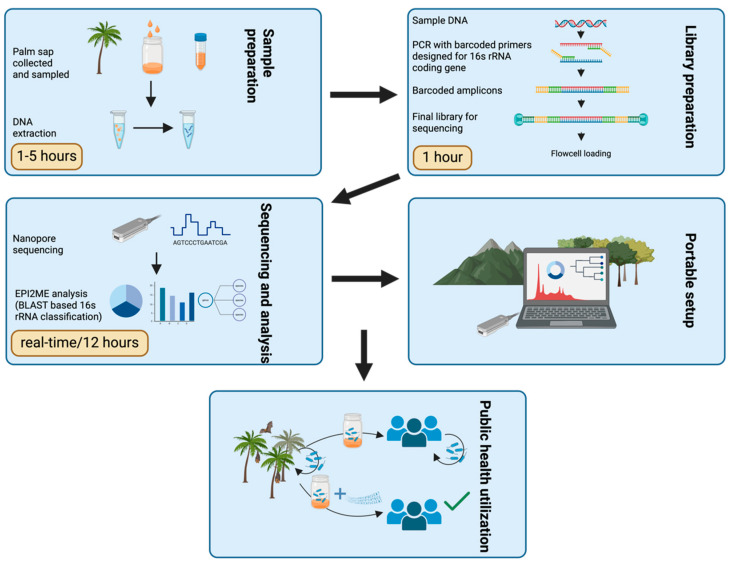
Representative scheme of the study workflow highlighting approximate processing times for each step.

**Figure 2 foods-13-01285-f002:**
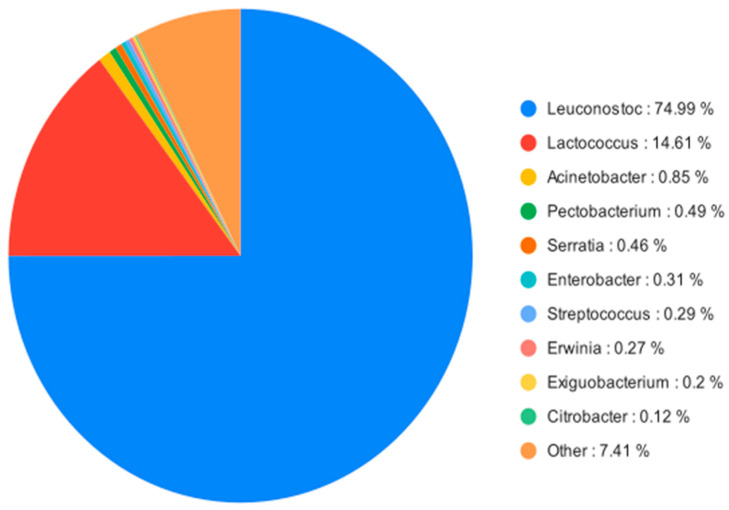
Ten most abundant bacterial genera from the composed dataset of all barcodes, represented as the relative abundance.

**Figure 3 foods-13-01285-f003:**
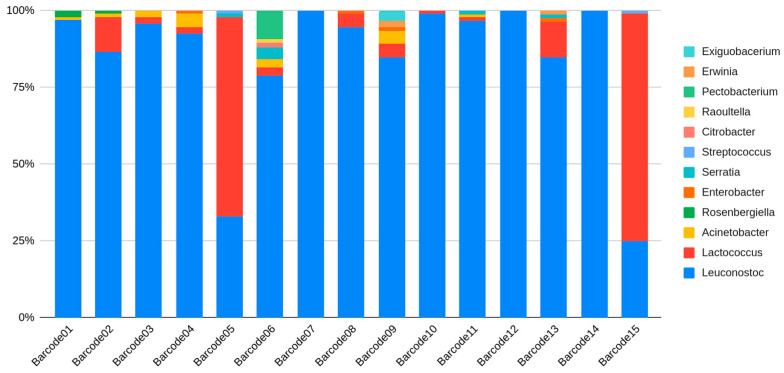
Bacterial composition at the genus level for all 15 investigated samples. Among the classified reads, those achieving a minimum of 1% are represented.

## Data Availability

The original contributions presented in the study are included in the article/Appendix A, further inquiries can be directed to the corresponding author.

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
