# Peer review of "Bacterial Metabarcoding of Raw Palm Sap Samples from Bangladesh with Nanopore Sequencing"

_foods, 2024, doi:10.3390/foods13091285_

Round 1
Reviewer 1 Report
Comments and Suggestions for Authors 1. The methodology of the paper is written in too general terms. 2. information on the exact origin of the samples is missing. When were they taken? 3. Line 108. instead of 94.98g/100g it should be 95g/100
Author Response
Thank you for the time and effort you have dedicated to reviewing and providing valuable feedback on our manuscript. Please find our responses below.
- The methodology of the paper is written in too general terms.
Response: Thank you for your useful comment, according to your request we extended the methodology part.
- information on the exact origin of the samples is missing. When were they taken? 3. Line 108. instead of 94.98g/100g it should be 95g/100
Response: Thank you, we corrected this part
Reviewer 2 Report
Comments and Suggestions for Authors
The manuscript entitled: “Bacterial metabarcoding of raw palm sap samples from Bangladesh with nanopore sequencing” is an extremely relevant manuscript, I congratulate the authors for their initiative.
Nevertheless, I must point out some pertinent issues, in my opinion:
1. Figure 2 does not depict any quantitative information.
2. Figure 3 is it possible to have an idea of the concentration of the microorganisms found? This would be highly relevant for their pathogenicity through ingestion. This would greatly improve the manuscript impact, in my opinion.
3. A more thorough description of the mechanisms of pathogenicity and symptoms of the most prevalent microorganisms would be greatly appreciated.
Minor amendments:
Please carefully revise the entire manuscript to ensure the adequate formatting of the genus and species names (particularly if they are italicized or not).
Author Response
Thank you for the time and effort you have dedicated to reviewing and providing valuable feedback on our manuscript. Please find our responses below.
The manuscript entitled: “Bacterial metabarcoding of raw palm sap samples from Bangladesh with nanopore sequencing” is an extremely relevant manuscript, I congratulate the authors for their initiative.
Nevertheless, I must point out some pertinent issues, in my opinion:
- Figure 2 does not depict any quantitative information.
Response: Thank you for your remark, we added the percentages of each component to the figure.
- Figure 3 is it possible to have an idea of the concentration of the microorganisms found? This would be highly relevant for their pathogenicity through ingestion. This would greatly improve the manuscript impact, in my opinion.
Response: We recognize the significance of dose dependency in infections. Nevertheless, it is important to note that the current manuscript presents a proof-of-concept study introducing a useful tool for food safety practices. A limitation of uncultured diagnostic methods, such as NGS, is their inability to perfectly gauge infectious dose. In our study, we provide relative abundance values and bacterial community compositions to offer a baseline understanding of this food product's nature. We have expanded the discussion in the manuscript to clarify these points for the readers.
- A more thorough description of the mechanisms of pathogenicity and symptoms of the most prevalent microorganisms would be greatly appreciated.
Response: Thank you for your comment, we clarified this part for the readers and describe the limitations and also the possibilities of our method in the MS. We added a paragraph regarding this.
Minor amendments:
Please carefully revise the entire manuscript to ensure the adequate formatting of the genus and species names (particularly if they are italicized or not)
Response: We corrected this throughout the manuscript
Reviewer 3 Report
Comments and Suggestions for Authors
The authors described the bacterial population in raw sap using point-of-collection barcoding techniques. The approach seems to be swift enough and provides somewhat useful information but the authors must help the reader exactly what the advantages of their approach are in terms of public health which, in my opinion, are not clear enough.
The report is not without some merit but I think is a quite simple approach even for a "Communication". Maybe the authors will want to mend their objective as something like "to describe the bacterial population in the food before processing".
Some specific comments:
Line 25 Please define NGS here
Line 31/45/101/103 please use italics as these are referring to the bacterial and bat genus. Please revise the rest of the manuscript for the correct use of italics in several scientific names
Line 45 Not all readers may be knowledgeable about the bat scientific name, maybe include a common name as well or a little description as to which bats are common in the area (e.g., flying fox)
Line 80 Please replace “falcon tubes”. Please indicate what polymer were the conical tubes made of, manufacturer? The material was frozen at what temperature?
Line 83 Please elaborate briefly on what exactly is the “manufacturer’s protocol”.
Line 84 Please replace uL with the Greek letter “Mu” μ.
Line 85/86 If this was done previously why not write them in that order? This seems a little confusing.
Line 89 Please separate the “1500” from its unit “bp”
Line 90/91/92 Again, elaborate briefly on the “Nanopore protocol” and “manufacturer’s protocol” and at least mention broadly the most relevant sections of the methodology.
Line 95 Which figures? Are we talking about figures like illustrations or figures as in numbers? This is a little confusing as well. Please rephrase this sentence to improve readability. Is the workflow mentioned here the same illustrated as in Figure 1? Not very clear
Line 97 Figure 1 is not mentioned in the text immediately before the figure appears.
Line 108 Please correct as “94.98 g/100 g”
Line 121 “antagonistic effect” against what? The idea here seems to be incomplete.
Line 124/125 Salmonella enterica var Thyphimurium or Enteritidis serovar has no italics and its first letter is written in uppercase.
Line 148 I consider Figure 2 is not inputting sufficient valuable information. I suggest improving it of removing it. Figure 3 gives similar information.
Line 151 “Overall, the species composition is similar through all 15 samples.” Please indicate at least the variation coefficient found.
My main concern about this study is the weight it has over food safety. Ok, so we already established that raw sap consumption may cause gastrointestinal issues. How is the barcoding addressing this issue exactly? How does it help? This should be addressed explicitly. Also, it is not very clear to me how does rapid response of barcoding at the site of collection is any different from sending these samples to the lab off-site. What benefit in terms of public health or response is this addressing?
Author Response
Thank you for the time and effort you have dedicated to reviewing and providing valuable feedback on our manuscript. Please find our responses below.
The authors described the bacterial population in raw sap using point-of-collection barcoding techniques. The approach seems to be swift enough and provides somewhat useful information but the authors must help the reader exactly what the advantages of their approach are in terms of public health which, in my opinion, are not clear enough.
The report is not without some merit but I think is a quite simple approach even for a "Communication". Maybe the authors will want to mend their objective as something like "to describe the bacterial population in the food before processing".
Some specific comments:
Line 25 Please define NGS here
Response: Corrected
Line 31/45/101/103 please use italics as these are referring to the bacterial and bat genus. Please revise the rest of the manuscript for the correct use of italics in several scientific names
Response: Corrected
Line 45 Not all readers may be knowledgeable about the bat scientific name, maybe include a common name as well or a little description as to which bats are common in the area (e.g., flying fox)
Response: Thank you for your useful remark, we extended this part as follows: „Flying foxes are represented by 51 species, mostly occurring in south-east Asia, South-Asia and Oceania. There is widespread habitat loss and hunting activities against these animals in most places within their distribution area. In South-Asia, mostly due to habitat loss and limited food sources, they are frequent visitors on agriculture-related fruits and sugar sources (such as the palm sap). Due to these contacts they are considered as priority species in relation to outbreak prevention strategies”
we added this reference: Breed AC, Field HE, Epstein JH, Daszak P. Emerging henipaviruses and flying foxes - Conservation and management perspectives. Biol Conserv. 2006 Aug;131(2):211-220. doi: 10.1016/j.biocon.2006.04.007. Epub 2006 Jun 6. PMID: 32226079; PMCID: PMC7096729.
Line 80 Please replace “falcon tubes”. Please indicate what polymer were the conical tubes made of, manufacturer? The material was frozen at what temperature?
Response: Thank you for your remark, we corrected this part and added more details.
Line 83 Please elaborate briefly on what exactly is the “manufacturer’s protocol”.
Response: Thank you, we extended the MS with brief explanations of the methods.
Line 84 Please replace uL with the Greek letter “Mu” μ.
Response: Corrected
Line 85/86 If this was done previously why not write them in that order? This seems a little confusing.
Response: Corrected
Line 89 Please separate the “1500” from its unit “bp”
Response: Corrected
Line 90/91/92 Again, elaborate briefly on the “Nanopore protocol” and “manufacturer’s protocol” and at least mention broadly the most relevant sections of the methodology.
Response: Corrected
Line 95 Which figures? Are we talking about figures like illustrations or figures as in numbers? This is a little confusing as well. Please rephrase this sentence to improve readability. Is the workflow mentioned here the same illustrated as in Figure 1? Not very clear
Response: Corrected
Line 97 Figure 1 is not mentioned in the text immediately before the figure appears.
Response: Corrected
Line 108 Please correct as “94.98 g/100 g”
Response: We corrected this part in accordance with Reviewer 1 comment.
Line 121 “antagonistic effect” against what? The idea here seems to be incomplete.
Response: Corrected
Line 124/125 Salmonella enterica var Thyphimurium or Enteritidis serovar has no italics and its first letter is written in uppercase.
Response: Corrected
Line 148 I consider Figure 2 is not inputting sufficient valuable information. I suggest improving it of removing it. Figure 3 gives similar information.
Response: Thank you for the useful remark, we updated the figure based on other reviewers requests.
Line 151 “Overall, the species composition is similar through all 15 samples.” Please indicate at least the variation coefficient found.
Response: Thank you for your useful comment. The nature of our study is descriptive, presenting a proof-of-concept workflow for enhancing food safety. As such, we did not include replicates or negative runs to derive exact statistical values; our goal was solely to understand the bacterial composition of this food product. We have made this clearer in the text.
My main concern about this study is the weight it has over food safety. Ok, so we already established that raw sap consumption may cause gastrointestinal issues. How is the barcoding addressing this issue exactly? How does it help? This should be addressed explicitly. Also, it is not very clear to me how does rapid response of barcoding at the site of collection is any different from sending these samples to the lab off-site. What benefit in terms of public health or response is this addressing?
Response: Thank you for your remark. In areas with limited infrastructure, maintaining a cold chain and transporting samples is a significant challenge. Logistical barriers can delay the timely delivery of samples, affect the acquisition of crucial diagnostic results, and hinder the implementation of intervention strategies where time is critical. In our study, we demonstrated an alternative strategy for implementing food safety diagnostic practices outside the traditional laboratory setting. While these methods are gaining attention in outbreak response strategies, as seen in the WHO guidelines (https://www.who.int/europe/publications/i/item/9789289054928), their potential in food safety has been less explored until now.
We added this paragraph regarding your concern to the conclusions part: „Logistical barriers can delay the timely delivery of samples, affect the acquisition of crucial diagnostic results, and hinder the implementation of intervention strategies where time is critical. In our study, we demonstrated an alternative strategy for implementing food safety diagnostic practices outside the traditional laboratory setting, using a simple NGS worklflow to overcome the diagnostic challenge of unculturable bacteria with classical methods.”
Reviewer 4 Report
Comments and Suggestions for Authors
Dear Editor
Many thanks for considering me a potential reviewer for the said article entitle; Bacterial metabarcoding of raw palm sap samples from Bangladesh with nanopore sequencing. This article is well structured and written, however, here are some queries and minor corrections (suggestions) that must be taken into consideration before the onward steps. My observations are as follow;
Major Comments
1. The whole article is poorly cited, for example, introduction is cited just 3 times (can’t imagine) and then authors had mention (under section Sequencing and Analysis) following the manufacturer's protocol…. where is the source/citation of protocol? I strongly recommend please do cite every information you are claiming.
2. The Authors need to italicized all scientific/generic names (throughout the whole manuscript.
3. Please re-write these sentences with more clearer objectives (The primary aim of this study was to demonstrate that nanopore sequencing-based bacterial metabarcoding is an effective and rapid method for identifying bacterial communities in food products, offering a significantly shorter diagnostic timeframe compared to conventional methods.).
Minor comments
1. Line-35 All scientific names (Leuconostoc species or Lactococcus species) should be italicized.
2. Abstract, what’s your conclusion and were your recommendations.
3. Figure 1; Sub-figure Library preparation, sample preparation, ‘s’ should be capitalized.
4. Figure 2. Please shows number and/or percentage of each species on their respective colors.
5. The authors number the section very nicely, but they didn’t do the same for sub-sections.
6. Also pay attention to the BACKSPACE usage, you did in the introduction, but did not follow it in the material method and ………
Comments on the Quality of English Language
Dear Authors
This article is well structured and written, however, I found that some sentences are soo longer please have pay attention that. Another technical issues I saw, please italize all scientific and generic names.
Thanks
Author Response
Thank you for the time and effort you have dedicated to reviewing and providing valuable feedback on our manuscript. Please find our responses below.
Many thanks for considering me a potential reviewer for the said article entitle; Bacterial metabarcoding of raw palm sap samples from Bangladesh with nanopore sequencing. This article is well structured and written, however, here are some queries and minor corrections (suggestions) that must be taken into consideration before the onward steps. My observations are as follow;
Major Comments
- The whole article is poorly cited, for example, introduction is cited just 3 times (can’t imagine) and then authors had mention (under section Sequencing and Analysis)following the manufacturer's protocol…. where is the source/citation of protocol? I strongly recommend please do cite every information you are claiming.
Response: Thank you for your remark, we added the requested references and additional citations.
- The Authors need to italicized all scientific/generic names (throughout the whole manuscript.
Response: Thank you for the useful comment, we corrected this mistake throughout the manuscript.
- Please re-write these sentences with more clearer objectives (The primary aim of this study was to demonstrate that nanopore sequencing-based bacterial metabarcoding is an effective and rapid method for identifying bacterial communities in food products, offering a significantly shorter diagnostic timeframe compared to conventional methods.).
Response: Thank you for your comment, we agree with your observation and highlighted the main idea, aims and conclusions in the introduction and conclusion parts.
Minor comments
- Line-35 All scientific names (Leuconostoc species or Lactococcus species) should be italicized.
Response: Corrected
- Abstract,what’s your conclusion and were your recommendations.
Response: Corrected
- Figure 1; Sub-figure Library preparation, sample preparation, ‘s’ should be capitalized.
Response: Corrected
- Figure 2. Please shows number and/or percentage of each species on their respective colors.
Response: Corrected
- The authors number the section very nicely, but they didn’t do the same for sub-sections.
Response: Corrected
- Also pay attention to the BACKSPACE usage, you did in the introduction, but did not follow it in the material method and ………
Response: Corrected
Round 2
Reviewer 3 Report
Comments and Suggestions for Authors
The authors tackled the majority of the comments in a satisfactory way.
Comments on the Quality of English Language
None
Reviewer 4 Report
Comments and Suggestions for Authors
Dear Authors,
I think, now the manuscript much refined,
However, English editing is needed.
Comments on the Quality of English Language
Dear Editor/authors
I think the manuscript fine, however, English editing is needed.
I have no objections, regarding the onward step.
Regards,
Asif